# Development and validation of a predictive model for critical illness in adult patients requiring hospitalization for COVID-19

Neha Paranjape[1]*, Lauren L. Staples[2], Christina Y. Stradwick[2], Herman Gene Ray[3], Ian J. Saldanha[4,5]

1 Department of Infectious Disease, Wellstar Medical Group, Marietta, Georgia, United States of America, 2 Analytics and Data Science Institute, Kennesaw State University, Marietta, Georgia, United States of America, 3 Analytics and Data Science Institute, Kennesaw State University, Marietta, Georgia, United States of America, 4 Center for Evidence Synthesis in Health, Department of Health Services, Policy, and Practice, Brown University School of Public Health, Providence, Rhode Island, United States of America, 5 Department of Epidemiology, Brown University School of Public Health, Providence, Rhode Island, United States of America

* neha.paranjape@wellstar.org

## Abstract

### Background

Identifying factors that can predict severe disease in patients needing hospitalization for COVID-19 is crucial for early recognition of patients at greatest risk.

### Objective

(1) Identify factors predicting intensive care unit (ICU) transfer and (2) develop a simple calculator for clinicians managing patients hospitalized with COVID-19.

### Methods

A total of 2,685 patients with laboratory-confirmed COVID-19 admitted to a large metropolitan health system in Georgia, USA between March and July 2020 were included in the study. Seventy-five percent of patients were included in the training dataset (admitted March 1 to July 10). Through multivariable logistic regression, we developed a prediction model (probability score) for ICU transfer. Then, we validated the model by estimating its performance accuracy (area under the curve [AUC]) using data from the remaining 25% of patients (admitted July 11 to July 31).

### Results

We included 2,014 and 671 patients in the training and validation datasets, respectively. Diabetes mellitus, coronary artery disease, chronic kidney disease, serum C-reactive protein, and serum lactate dehydrogenase were identified as significant risk factors for ICU transfer, and a prediction model was developed. The AUC was 0.752 for the training dataset and 0.769 for the validation dataset. We developed a free, web-based calculator to facilitate use of the prediction model (https://icucovid19.shinyapps.io/ICUCOVID19/).

**Data Availability Statement:** All relevant data are within the manuscript and its Supporting Information files.

**Funding:** The authors received no specific funding for this work.

**Competing interests:** The authors have declared that no competing interests exist.

## Conclusion

Our validated, simple, and accessible prediction model and web-based calculator for ICU transfer may be useful in assisting healthcare providers in identifying hospitalized patients with COVID-19 who are at high risk for clinical deterioration. Triage of such patients for early aggressive treatment can impact clinical outcomes for this potentially deadly disease.

## Introduction

COVID-19 is a disease cause by SARS-CoV2, a novel coronavirus first identified in Wuhan, China in December 2019. Since then, it has spread globally resulting in an ongoing pandemic. The United States (U.S.) has been an epicenter for many months and, as of March 2021, has had over 29 million confirmed cases, with over 520,000 deaths [1].

The clinical spectrum of COVID-19 varies from a mild upper respiratory tract infection to severe life-threatening respiratory failure. Although several risk factors, such as age, sex, and certain co-morbidities, have been identified, it remains unclear why some patients recover well within a few days of hospitalization but others progress to a critical illness. While there are several ongoing clinical trials evaluating therapeutic agents for COVID-19, identifying the subset of patients that would benefit the most from these therapies remains a challenge. This is especially true during times of a case surge and/or drug shortage. Studies from China and Italy have demonstrated that some factors, such as age, comorbidities, race, and certain laboratory markers, are associated with risk of developing severe disease [2, 3]. The Modified Early Warning Score (MEWS) for clinical deterioration that was developed in 2015 has not shown to be accurate in predicting the need for intensive care in patients with COVID-19 [4]. Other predictive models have been developed for hospitalized COVID-19 patients in China and U.S [5–9]. However, key differences in population demographics between China and the U.S., and among states in the U.S., indicated the need for development and validation of a prognostic predictive model for our center. Identifying factors that can predict severe disease in hospitalized patients with COVID-19 is crucial for early recognition of at-risk individuals and for informing policy decisions regarding triage of patients who would benefit most from early treatment.

This paper describes the development and validation of a model and simple online calculator to predict ICU admission in hospitalized patients with COVID-19.

## Methods

We developed and validated a prediction model based on analysis of a retrospective sample of patients admitted to a large metropolitan health care system consisting of nine community hospitals in the state of Georgia, U.S. The study was granted exempt status and the requirement for obtaining informed consent was waived by the Wellstar Health System Institutional Review Board (Approval Number: 1611062–1). The reporting of this study adheres to the Transparent Reporting of a Multivariable Prediction Model for Individual Prognosis or Diagnosis (TRIPOD) Statement [10].

### Training and validation datasets

We analyzed de-identified data on a convenience sample of all adult patients (18 years or older) who were confirmed positive for SARS-CoV2 via a nasopharyngeal swab polymerase

chain reaction (PCR) test and hospitalized with COVID-19 between March 1, 2020 and July 31, 2020. We excluded patients admitted directly to the ICU because they already experienced our outcome of interest (i.e., ICU transfer) at their earliest observed time-point.

Patients admitted from March 1 through July 10, 2020 (75% of the sample) were included in the training dataset, while those admitted from July 11 through July 31, 2020 (25% of the sample) were included in the validation dataset. The cut-point determining the training/validation split was selected to provide a 75%:25% training/validation split, while maintaining temporal order.

## Data extracted and variables considered

We extracted all data from electronic medical records. Variables gathered included:

- Demographics, e.g., age, sex, race, body mass index (BMI);

- Temperature and oxygen saturation on room air (RA) on admission;

- Laboratory values of C-reactive protein (CRP), lactate dehydrogenase (LDH), ferritin, D-dimer, and absolute lymphocyte count on admission;

- Presence of comorbidities, e.g., hypertension (HTN), diabetes mellitus (DM), chronic kidney disease (CKD), asthma and chronic obstructive pulmonary disease (COPD), coronary artery disease (CAD); and

- Status regarding transfer to the ICU service (irrespective of the patient's physical location).

The criteria for transfer to the ICU service remained the same as that for non-COVID patients. ICU service transfer was initiated for patients requiring higher level of care as deemed necessary by the on-call critical care medicine provider.

We used the above pool of variables to develop a simple and easy to use tool for clinicians based on data commonly available at most acute care facilities, including small community or rural hospitals. For patients with multiple hospitalizations, we considered their first hospitalization for this study.

## Statistical analysis

We conducted bivariable and multivariable analyses of the associations between various independent variables and the dependent variable of interest (transfer to ICU service). Chi-square tests and simple regressions were used to provide descriptive statistics. For continuous data, such as laboratory values, outliers exceeding three times the standard deviation ($3\sigma$) were replaced with $3\sigma$. The maximum number of outliers for each continuous variable remained low (<1.4%). Because the amount of missingness of continuous variables was low (<10%; see S1 Appendix), missing data were imputed using median values. There were no missing data for sex or race. Each comorbidity was coded as "1 = present" or "0 = absent".

We conducted a multivariable logistic regression to determine the probability (from the odds) of being transferred to the ICU service, with the goal of providing a model for predicting the probability of ICU transfer for future cases. We used a backwards selection stepwise method, with a P value threshold of 0.05. To validate the model, we graphed a receiver operating characteristic (ROC) curve and calculated the area under the curve (AUC). We also calculated the sensitivity, specificity, positive predictive value, negative predictive value, and likelihood ratio.

We used Python® version 3.7.6 for preprocessing and SAS® server version 9.4 for Chi-square tests and regression analyses [11, 12].

### IRB approval

Approval Number: 1611062–1

## Results

### Description of study population (combined sample)

Among all 2,685 eligible patients, the mean age was 59.7 years (standard deviation [SD] = 17.3) [Table 1]. Approximately half of the patients (49%) were of female sex. The mean BMI was 31.7 kg/m$^2$ (SD = 8.9). Forty-seven percent of patients were Black and 34% were White. HTN (72%) and DM (55%) were the most common comorbidities.

**Training dataset.** The demographic characteristics and comorbidities of the 2,014 patients in the training dataset, which comprised 75% of the combined sample, were very similar to that of the combined sample [Table 1].

**Validation dataset.** The mean age was slightly lower (57.8 years) in this group of 671 patients [Table 1]. Forty-two percent of patients were Black and 35% were White. The

**Table 1. Demographic characteristic and comorbidities for patients in the training dataset (March 1 to July 10, 2020) and validation dataset (July 11 to July 31, 2020).**

| Characteristic | | Training Dataset (N = 2,014) | | Validation Dataset (N = 671) | | Combined Sample (N = 2,685) | |
|---|---|---|---|---|---|---|---|
| | | n | (%) | n | (%) | N | (%) |
| *Age* (years) | | | | | | | |
| | Mean | 60.3 | | 57.8 | | 59.7 | |
| | Standard deviation | 17.4 | | 16.8 | | 17.3 | |
| | Minimum | 18 | | 19 | | 18 | |
| | Maximum | 104 | | 101 | | 104 | |
| *Sex* n (%) | | | | | | | |
| | Female | 990 | (49) | 315 | (47) | 1305 | (49) |
| | Male | 1024 | (51) | 356 | (53) | 1380 | (51) |
| *Body mass index (BMI)* | | | | | | | |
| | Mean | 31.5 | | 32.4 | | 31.7 | |
| | Standard deviation | 8.9 | | 8.7 | | 8.9 | |
| | Minimum | 12.8 | | 12.8* | | 12.8 | |
| | Maximum | 59.6 | | 59.6* | | 59.6 | |
| *Race* n (%) | | | | | | | |
| | Black/African American | 981 | (49) | 281 | (42) | 1262 | (47) |
| | White/Caucasian | 667 | (33) | 236 | (35) | 1262 | (34) |
| | Other | 366 | (18) | 153 | (23) | 520 | (19) |
| *Comorbidities* n (%) | | | | | | | |
| | Hypertension (HTN) | 1495 | (74) | 437 | (65) | 1932 | (72) |
| | Diabetes mellitus (DM) | 1118 | (56) | 354 | (53) | 1472 | (55) |
| | Coronary artery disease (CAD) | 620 | (31) | 155 | (23) | 775 | (29) |
| | Chronic kidney disease (CKD) | 523 | (26) | 123 | (18) | 646 | (24) |
| | Malignancies | 270 | (13) | 63 | (9) | 333 | (12) |
| | Chronic obstructive pulmonary disease (COPD)/Asthma | 67 | (3) | 0 | (0) | 67 | (2) |

* Validation data values have been winsorized to the minimum/maximum values of the training dataset so as not to extrapolate.

proportion of patients with each of the comorbid conditions was slightly lower in the validation dataset than the training and combined datasets.

**Clinical and laboratory values and ICU transfer status (all datasets).** Mean patient temperature, oxygen saturation, CRP level, and absolute lymphocyte count were similar across datasets [Table 2]. Compared with patients in the training dataset, patients in the validation dataset had somewhat higher levels of LDH, but somewhat lower levels of ferritin and D-dimer. In the combined sample, 37% of patients were transferred to the ICU. This percentage was 40% and 29% in the training and validation datasets, respectively.

**ICU risk score model development.** In multivariable logistic regression analyses of the training dataset (2,014 patients), the following variables were predictive of ICU transfer [Table 3]:

- CAD, CKD, and DM (comorbidities): The odds of ICU transfer were higher by 32% in patients with CAD, 59% in patients with CKD, and 97% in patients with DM.

- Serum CRP: A 1-unit increase in serum CRP was associated with a 5.4% higher odds of ICU transfer.

- Serum LDH: A 1-unit increase in serum LDH was associated with a 0.4% higher odds of ICU transfer.

**ICU risk score model validation.** We validated the model using data from the 671 patients in the validation dataset. As seen in Fig 1, the model performed well in both the

**Table 2. Clinical and laboratory values at admission and ICU transfer status for patients in the training dataset (March 1 to July 10, 2020) and validation dataset (July 11 to July 31, 2020).**

| Characteristic | | Training Dataset (N = 2,014) | | Validation Dataset (N = 671) | | Combined Sample (N = 2,685) | |
|---|---|---|---|---|---|---|---|
| | | n | (%) | n | (%) | N | (%) |
| *Temperature (F)* | | | | | | | |
| | Mean | 98.8 | | 98.6 | | 98.8 | |
| | Standard deviation | 1.3 | | 1.2 | | 1.3 | |
| | Minimum | 94.7 | | 94.7* | | 94.7 | |
| | Maximum | 103.0 | | 103.0* | | 103.0 | |
| *Oxygen saturation (%)* | | | | | | | |
| | Mean | 95.0 | | 94.6 | | 94.9 | |
| | Standard deviation | 3.2 | | 3.1 | | 3.2 | |
| | Minimum | 84.4 | | 84.4* | | 84.4 | |
| | Maximum | 100 | | 100 | | 100 | |
| Laboratory values at admission (mean (SD)) | | | | | | | |
| | C-reactive protein (mg/dl) | 9.8 (8.9) | | 9.9 (8.7) | | 9.8 (8.9) | |
| | Lactate dehydrogenase (mg/ml) | 371.7 (196.1) | | 384.1 (195.7) | | 374.9 (196.0) | |
| | Ferritin (ng/dl) | 1033.2 (1895.5) | | 948.5 (1495.5) | | 1011.8 (1795.3) | |
| | D-dimer (ng/dl) | 1587.0 (3209.2) | | 1128.0 (2829.6) | | 1462.7 (3117.1) | |
| | Absolute lymphocyte count ($10^9$/dl) | 1.2 (0.8) | | 1.1 (0.6) | | 1.2 (0.7) | |
| *Transferred to intensive care unit (ICU)* | | | | | | | |
| | No | 1210 | (60) | 479 | (71) | 1689 | (63) |
| | Yes | 804 | (40) | 192 | (29) | 996 | (37) |

* Validation data values have been winsorized to the minimum/maximum values of the training dataset so as not to extrapolate.

**Table 3. Final model from stepwise backwards selection, multivariate logistic regression predicting intensive care unit status.**

| Variable | Estimate | Odds Ratio (95% CI) | p-Value |
|---|---|---|---|
| Intercept | -2.9826 | 0.0507 | <0.0001 |
| Coronary artery disease (CAD) | 0.2774 | 1.3197 (1.055, 1.65) | 0.0149 |
| Chronic kidney disease (CKD) | 0.4636 | 1.5898 (1.259, 2.007) | <0.0001 |
| Diabetes mellitus (DM) | 0.6756 | 1.9652 (1.599, 2.415) | <0.0001 |
| Serum C-reactive protein (CRP) | 0.0525 | 1.0539 (1.041, 1.067) | <0.0001 |
| Serum lactate dehydrogenase (LDH) | 0.00397 | 1.0040 (1.003, 1.005) | <0.0001 |

All model inputs into stepwise selection are listed in the S1 Appendix.

training and validation datasets, with AUC values of 0.752 and 0.769, respectively. At a probability cut-off of 20%, our model correctly predicted 95% of actual ICU transfers (PPV of 34%) and 27% of non-ICU transfers (NPV of 94%). The Likelihood Ratio for positive and negative prediction was 1.31 and 0.17 respectively. (Appendix E in S1 Appendix)

Based on the predictive model developed in this study, we designed a free, easy to use, online web-based calculator, to help clinicians predict the probability of hospitalized patients with COVID-19 being transferred to the ICU (https://icucovid19.shinyapps.io/ICUCOVID19/).

## Discussion

### Summary of findings

We developed and validated a prediction model in a total of 2,685 patients hospitalized with COVID-19 to help clinicians identify patients who, at admission, are at risk for subsequent clinical deterioration, thus potentially helping appropriate triage and management of these patients. The model was developed quickly in response to local need during the COVID-19 surge when hospitals in our health system were at crisis capacity. The potential value of this model is enhanced by the fact that its required inputs (DM, CAD, CKD, serum LDH, and serum CRP) are commonly available at most acute care facilities, including small community or rural hospitals. Measurements of these inputs are already routine in patients hospitalized with COVID-19. Our risk calculator was developed using variables collected at admission and is therefore is a true predictor of severe disease in absence of in-patient treatments, such as remdesivir and dexamethasone, that are now considered standard of care for hospitalized patients.

The ability to predict with reasonable accuracy which patients are most likely to deteriorate and need ICU care can be particularly useful when a hospital is at crisis capacity. In such situations of bed shortages, clinicians often need to prioritize the sickest patients for hospitalization. Some patients who would otherwise be hospitalized may be denied admission. The model and risk calculator we developed in the current study could greatly help decision-making in this context. Along with the clinicians' judgment, patients with an ICU transfer probability score of 20% or higher may be given priority for in-patient treatment over admission to an observation unit or to be sent home.

### Comparison with similar studies

While several other comorbid conditions have been identified as risk factors for severe COVID-19, our study uncovers DM, CAD, and CKD as the most significant factors [13, 14]. In addition to the easy availability of the variables that serve as inputs, our model's simplicity is one of its greatest strengths. Using fewer variables, our model has similar performance (in terms of AUC) as another predictive model published in October 2020 [9].

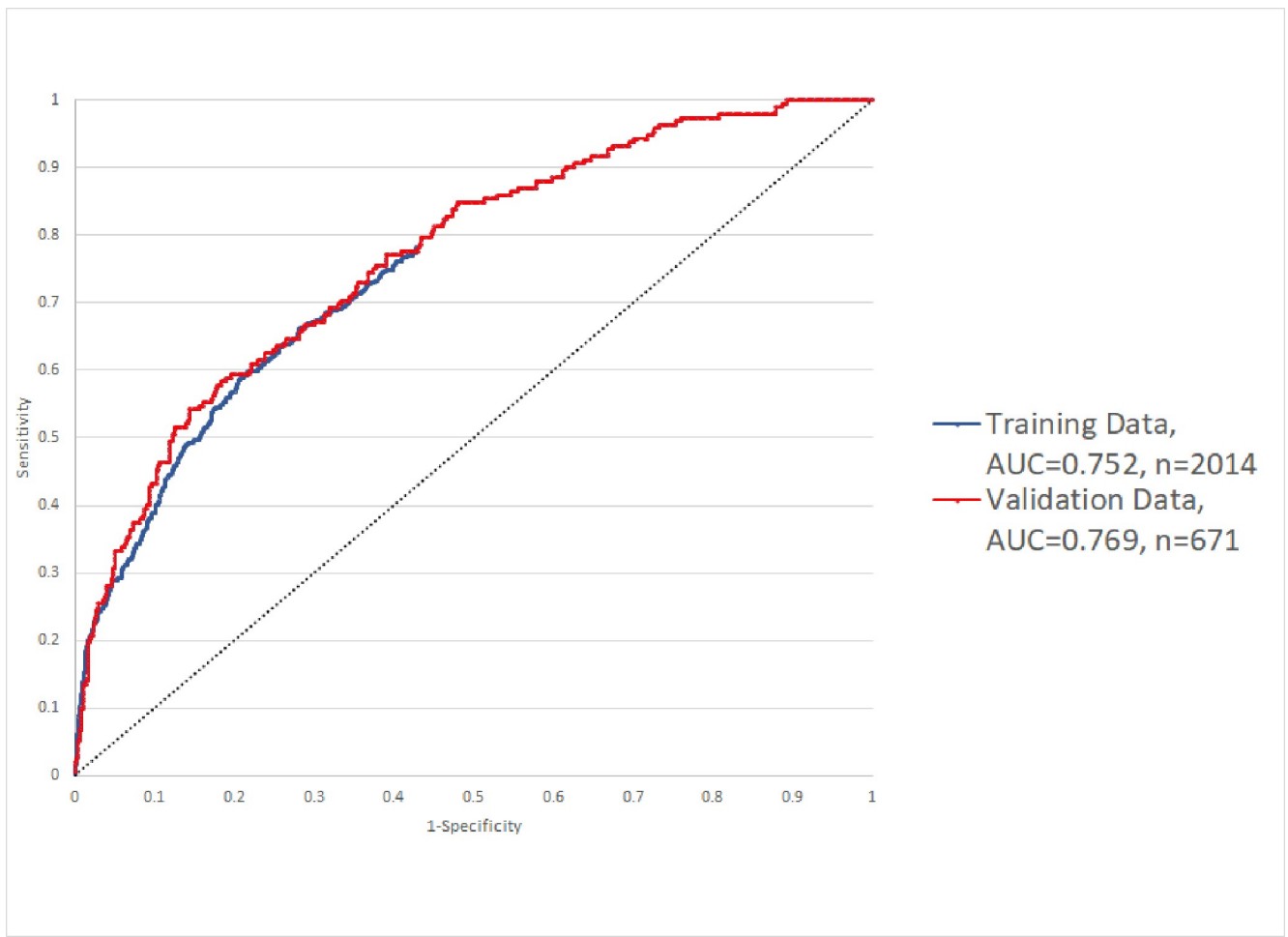

**Fig 1. Receiver Operator Characteristic (ROC) curve in the training and validation datasets.**

Our health system serves a metropolitan population in Georgia with significant chronic conditions and health disparities [15]. Although age and high BMI are described as a risk factor for COVID-19 infection in the general population, these did not emerge as independent risk factors in our analysis [16, 17]. This was likely because our study included only hospitalized patients and a majority of patients needing hospitalization were older (mean age of 62 years) and had a higher BMI (mean 31 kg/m$^2$).

LDH is an intracellular enzyme that catalyzes the conversion of lactate to pyruvate. It is found in most organ tissue cells and is known to be a marker of cell injury. Elevated levels of LDH have been detected in patients with COVID-19, and it has also been identified, as an independent predictor of disease severity [18]. CRP is an acute phase protein of hepatic origin that binds to dead or dying cells and activates the complement system, promoting phagocytosis by macrophages. This biomarker has also been shown to predict disease severity [19].

## Strengths and limitations

Our study has a number of strengths. *First*, our predictive model included a limited number of variables that are easily available at hospital presentation, including small tertiary care hospitals. *Second*, we had a large sample size in both the training as well as validation datasets.

*Third*, the demographics of patients in the current study are highly representative of hospitalized patients in the U.S., which are more diverse and include more vulnerable populations than elsewhere [20]. Further, our health system includes nine community hospitals with a total capacity of over 2,500 beds. The model we developed may therefore be generalizable to other similar acute care facilities in the U.S. *Fourth*, apart from laboratory values, our analysis included presence of co-morbidities such as DM, CAD, and CKD that have been shown to be risk factors for severe disease [13, 14]. *Fifth*, we defined the need for ICU as the transfer to ICU service rather than physical location of the patient. Due to bed shortages during the height of the pandemic, hospitals instituted makeshift locations for ICU-level care. We were thus able to accurately capture true ICU transfers. *Lastly*, we created an easy, ready to use, web-based calculator, freely accessible to clinicians.

Our study also has some limitations. *First*, we developed and validated the prediction model only for hospitalized patients. Thus, the model may not be applicable to non-hospitalized patients with COVID-19. *Second*, although it involved a large sample size, this study is based on a single health system. ICU transferal practices may differ across health systems. *Third*, we conducted data extraction from electronic medical records only. The accuracy of patient demographics and pre-existing conditions could not be independently verified. *Fourth*, of the laboratory values, three the variables (CRP, LDH, and D-dimer) had <10% missing values that were imputed by the median. This approach was similar to a recent predictive model published in October 2020 [9]. *Fifth*, due to the constant evolving nature of COVID-19 research we acknowledge that we included only a finite number of variables and characteristics that were implicated in prediction of severe disease during the initial months of the pandemic. *Lastly*, we recognize that this tool may not be suitable for patient populations with significantly different genetic and socioeconomic backgrounds from ours, however similar site-specific models can be developed quickly and provide a useful tool in triage of patients during surge crises, as many centers are currently experiencing.

In summary, we developed and validated a prognostic model to predict subsequent ICU transfer in hospitalized patients with COVID-19. We do not suggest that the decision for patient transfer to the ICU be based solely on the prediction model described in the current study. The model should be used in conjunction with the treating clinical team's evaluation in the context of all available information and history regarding an individual patient.

## Supporting information

**S1 Appendix.**
(DOCX)

**S1 Dataset.**
(XLSX)

**S2 Dataset.**
(XLSX)

## Author Contributions

**Conceptualization:** Neha Paranjape.

**Data curation:** Neha Paranjape.

**Formal analysis:** Lauren L. Staples, Herman Gene Ray.

**Investigation:** Neha Paranjape.

**Methodology:** Neha Paranjape, Lauren L. Staples, Herman Gene Ray.

**Project administration:** Neha Paranjape.

**Software:** Christina Y. Stradwick.

**Supervision:** Neha Paranjape, Herman Gene Ray, Ian J. Saldanha.

**Validation:** Lauren L. Staples, Herman Gene Ray.

**Visualization:** Neha Paranjape.

**Writing – original draft:** Neha Paranjape, Lauren L. Staples.

**Writing – review & editing:** Neha Paranjape, Lauren L. Staples, Ian J. Saldanha.

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
