## [Decision Letter · Decision Letter 0]

23 Feb 2021

PONE-D-21-02303

Development and validation of a predictive model for critical illness in adult patients requiring hospitalization for COVID-19

PLOS ONE

Dear Dr. Paranjape,

Thank you for submitting your manuscript to PLOS ONE. After careful consideration, we feel that it has merit but does not fully meet PLOS ONE’s publication criteria as it currently stands. Therefore, we invite you to submit a revised version of the manuscript that addresses the points raised during the review process.

We look forward to receiving your revised manuscript.

Kind regards,

Aleksandar R. Zivkovic

Academic Editor

PLOS ONE

Reviewers' comments:

Reviewer's Responses to Questions

**Comments to the Author**

1. Is the manuscript technically sound, and do the data support the conclusions?

Reviewer #1: Yes

Reviewer #2: Yes

2. Has the statistical analysis been performed appropriately and rigorously? 

Reviewer #1: Yes

Reviewer #2: Yes

3. Have the authors made all data underlying the findings in their manuscript fully available?

Reviewer #1: Yes

Reviewer #2: No

4. Is the manuscript presented in an intelligible fashion and written in standard English?

Reviewer #1: Yes

Reviewer #2: Yes

5. Review Comments to the Author

Reviewer #1: This manuscript is very clearly written and the objective is very well defined: to set a predicted risk score for ICU admission. I would say that external validity for this score is questionable, but authors have defined that the goal is to build a predictive score for a defined population. In this sense, I think statistical analysis was appropriate. I appreciate the low amount of missing data in such a large population.

My concerns are quoted by authors in the discussion: the comorbidities the authors found to be related with ICU admission in this population are not the ones we have seen to be ICU admission predictors in literature so far. I don't understand why authors have chosen only oxygen saturation to account for respiratory pattern; I would also include respiratory frequency and need for any ventilatory support out of the ICU - these are main predictors of need for intensive care and mechanical ventilation. It's not clear whether patients were in room air or if they are already using some kind of oxygen support. In addition, the authors present median BMI, but don't account for BMI in their score - this is strange, since high BMI is a predictor of ICU admission for COVID-19 in many populations across the world.

I would also ask for data regarding patients who were and were not admitted to the ICU. A table comparing data (such as presented in tables 1 and 2) between those admitted vs. not admitted to the ICU is key. I understand that CKD, CAD, CRP and LDH are predictors of ICU admission in this sample, but I would like to know median values for each one in those admitted vs. not admitted to ICU.

Reviewer #2: Thanks for your invitation. Prediction for severe or critical illness of COVID-19 is an important issue which are attracting increasing attention for clinicians and researchers. The authors developed and validated a simple and easily available tool to predict ICU service for in-hospital patients with COVID-19. Followings are the comments to the authors that should be considered.

1. There are emerging articles reported the predictors or prognostic models for severe or critical illness in patients with COVID-19. Did you systematically summarize the reported predictors or variables that should be included in your prediction model? I am afraid a wide array of predictors, such as neutrophil to lymphocyte ratio, blood glucose level, serum creatinine, procalcitonin, troponin and so on, were missed in your model.1-3 In addition, many of them are commonly available at most acute care facilities. So why those potential predictors were not considered in your model?

2. What is the definition and method for measurement of outcome to be predicted? What is the criteria of patients transfer to ICU service?

3. This is a retrospective study based on data from health care system. So there is more potential risk of bias compared with perspective study. What methods have you used to minimize the risk of bias in the present study?

4. I noted that you had compared your work with similar studies in discussion section. Many large scale, high-quality articles had published in top journals, and more and more similar articles are emerging. So, except for easy availability and feasibility, what are the main advantages of your model compared with other prediction model? What’s new in your work?

5. This is a single center study. There was no external validation. The prediction model is feasible and useful for local need. However, the generalizability of this model to external population might be unclear. Did you have any plan to validate your model in external data?

6. The present study did not mention treatments patients received. Some of them may be protective factor in your model.

7. Except for logistic regression method, it will be better to compare the accuracy with other multivariable models.

8. Some references are incomplete. (line 344-346, 358-370)

REFERENCES

[1] Petrilli C M, Jones S A, Yang J, et al. Factors associated with hospital admission and critical illness among 5279 people with coronavirus disease 2019 in New York City: prospective cohort study[J]. Bmj, 2020, 369.

[2] Zhu L, She Z G, Cheng X, et al. Association of blood glucose control and outcomes in patients with COVID-19 and pre-existing type 2 diabetes[J]. Cell metabolism, 2020, 31(6): 1068-1077. e3.

[3] Liang W, Liang H, Ou L, et al. Development and validation of a clinical risk score to predict the occurrence of critical illness in hospitalized patients with COVID-19[J]. JAMA internal medicine, 2020, 180(8): 1081-1089.

6. PLOS authors have the option to publish the peer review history of their article (what does this mean?). If published, this will include your full peer review and any attached files.

Reviewer #1: **Yes: **Tarsila Vieceli

Reviewer #2: No

---

## [Author Response · Author response to Decision Letter 0]

6 Mar 2021

Please see attached Response to Reviewers and Revised Manuscript

---

## [Editor Report · Decision Letter 1]

8 Mar 2021

Development and validation of a predictive model for critical illness in adult patients requiring hospitalization for COVID-19

PONE-D-21-02303R1

Dear Dr. Paranjape,

We’re pleased to inform you that your manuscript has been judged scientifically suitable for publication and will be formally accepted for publication once it meets all outstanding technical requirements.

Kind regards,

Aleksandar R. Zivkovic

Academic Editor

PLOS ONE

---

## [Editor Report · Acceptance letter]

11 Mar 2021

PONE-D-21-02303R1 

Development and validation of a predictive model for critical illness in adult patients requiring hospitalization for COVID-19 

Dear Dr. Paranjape:

I'm pleased to inform you that your manuscript has been deemed suitable for publication in PLOS ONE. Congratulations! Your manuscript is now with our production department. 

Kind regards, 

on behalf of

Dr. Aleksandar R. Zivkovic 

Academic Editor

PLOS ONE